# Wavelet Ridges in EEG Diagnostic Features Extraction: Epilepsy Long-Time Monitoring and Rehabilitation after Traumatic Brain Injury [note 1]

**DOI:** 10.3390/s21185989

**Published:** 2021-09-07

**Authors:** Yury Vladimirovich Obukhov, Ivan Andreevich Kershner, Renata Alekseevna Tolmacheva, Mikhail Vladimirovich Sinkin, Ludmila Alekseevna Zhavoronkova

**Affiliations:** 1Kotelnikov Institute of Radio Engineering and Electronics of RAS, Mokhovaya St. 11-7, 125009 Moscow, Russia; yuvobukhov@mail.ru (Y.V.O.); tolmatcheva@yandex.ru (R.A.T.); 2Department of Neurosurgery of the Sklifosovsky Research Institute for Emergency Medicine of Moscow Healthcare Department, Bolshaya Sukharevskaya Square 3, 129090 Moscow, Russia; sinkinmv@sklif.mos.ru or; 3Laboratory of Invasive Neurointerfaces of the Research Institute TechnoBioMed, A.I. Yevdokimov Moscow State University of Medicine and Dentistry, Delegatskaya St. 20 p.1, 127473 Moscow, Russia; 4Laboratory of General and Clinical Neurophysiology of the Institute of Higher Nervous Activity and Neurophysiology of RAS, Butlerova St. 5a, 117485 Moscow, Russia; lzhavoronkova@hotmail.com or

**Keywords:** electroencephalogram, wavelet spectrum, ridge, segmentation, phase connectivity, epilepsy, traumatic brain injury

## Abstract

Interchannel EEG synchronization, as well as its violation, is an important diagnostic sign of a number of diseases. In particular, during an epileptic seizure, such synchronization occurs starting from some pairs of channels up to many pairs in a generalized seizure. Additionally, for example, after traumatic brain injury, the destruction of interneuronal connections occurs, which leads to a violation of interchannel synchronization when performing motor or cognitive tests. Within the framework of a unified approach to the analysis of interchannel EEG synchronization using the ridges of wavelet spectra, two problems were solved. First, the segmentation of the initial data of long-term monitoring of scalp EEG with various artifacts into fragments suspicious of epileptic seizures in order to reduce the total duration of the fragments analyzed by the doctor. Second, assessments of recovery after rehabilitation of cognitive functions in patients with moderate traumatic brain injury. In the first task, the initial EEG was segmented into fragments in which at least two channels were synchronized, and by the adaptive threshold method into fragments with a high value of the EEG power spectral density. Overlapping in time synchronized fragments with fragments of high spectral power density was determined. As a result, the total duration of the fragments for analysis by the doctor was reduced by more than 60 times. In the second task, the network of phase-related EEG channels was determined during the cognitive test before and after rehabilitation. Calculation-logical and spatial-pattern cognitive tests were used. The positive dynamics of rehabilitation was determined during the initialization of interhemispheric connections and connections in the frontal cortex of the brain.

## 1. Introduction

Wavelet transform (WT) is widely used in the processing and analysis of non-stationary signals [1,2,3,4,5]. Since the 1990s, in various fields of biology and medicine [6], in neurophysiology [7], discrete and continuous wavelet transforms have been used to extract diagnostic information from signals and images of various types.

Considering EEG as a simultaneously amplitude and phase modulated analytical signal, and if the scanning wavelet width is narrower than the changes in signal phase, then it is possible to use the property of the wavelet spectrum ridge, namely those that the amplitude and phase of the signal are equal to the amplitude and phase of the wavelet spectrum ridge [8,9,10,11]. Thus, defining the ridge as the absolute maximum of the wavelet spectrum at each moment (reference point) of time, we obtain instantaneous values of the amplitude, frequency and phase of the signal. This very useful property of WT ridges makes it easy to find interchannel synchronized EEG fragments during epileptic seizures (ESs) in long-term clinical monitoring data, restoring the cognitive functions of patients after moderate traumatic brain injury using interchannel phase coupling link analysis, and other tasks of EEG diagnostics.

WT is used for EEG decomposition into time-frequency fragments for the subsequent detection of epileptic seizures (ESs). Currently, there are many publications on the use of various classifiers for the detection and prediction of an epileptic seizure in EEG signals using various classifiers [12,13,14,15,16,17,18,19]. Initial data on epilepsy monitoring should be preliminarily processed, including removal of artifacts and filtering noise to get a clean epilepsy EEG signal for the next step, feature extraction and classification [18,19].

In decision support systems, methods based on the analysis of EEG patterns are most often used and one of them is the “Persyst” system by Persyst Development Corpo ation (https://www.persyst.com, accessed on 13 August 2021). To detect ES in the time domain, discrete-time sequences are analyzed into which the original EEG signal is divided. One of such methods is based on tracking successive extrema in the selected time interval of the signal and evaluating the histogram of the amplitude difference and time separation between the maximum and minimum values of the histogram [20]. The different approaches for detecting ESs which were proposed in the time domain are the calculation of signal energy [21]; the frequency characteristics of the signal were studied: the index of the phase slope of multichannel EEG [22]; frequency-moment signatures [23]; entropy features [24]; Bayesian linear discriminant analyses of lacunarity and fluctuation index [25], four-level Daubechies wavelet transform [26] and five-level wavelet decomposition method [27]. The most promising method of EEG analysis is the study of the parameters of the ridges of wavelet spectrograms. In the EEG signals for the detection of epileptic seizures, the dynamics of synchronization and changes in the phase ratio before, during and after the seizures are monitored.

At present, attempts are being made to improve methods for detecting ESs in EEG. Paper [28] describes a way to improve the support vector machine method by adding an adaptive median feature baseline correction method. A combination of methods is also used to search for ESs, for example, complementary ensemble empirical mode decomposition with extreme gradient boosting [15]. A method has been proposed that combines time-domain feature analysis and entropy calculation [16]. A similar combination was also presented in work [14], but the study of parameters in the time domain was used to segment the signal sections, in order to then carry out analysis using machine learning methods. To differentiate ESs from non-seizure events, neural networks [13] and similar methods are used, such as the method of binarization of frequency and temporal features of signal fragments [29].

It should be noted that the estimation of the accuracy and specificity of the classification was carried out on EEG fragments previously selected and annotated by EEG neurosciensists as ictal and interictal events. The most representative databases are the EPILEPSIAE database [30], the Temple University Hospital EEG Data Corpus [31], Bonn epilepsy dataset etc.

Clinical EEG investigations of epilepsy consist in long-term (several days) monitoring of multichannel EEG using scalp or intracrinial electrodes in the presence of various artifacts: the electrical activity that is not recorded in the cerebral zone, such as that due to the equipment, patient behavior or the environment; eye movement and chewing are common events that can often be confused with a spike; signals instrument fluctuations and artifacts of vital activity [32]. It can be seen from the review that the methods for removing artifacts described in the literature are mainly focused on removing one type of possible artifacts or ocular and muscular ones present in the real initial data of long-term EEG monitoring. In general, we can conclude that the problem of automated removal of artifacts of various types from the initial data of long-term EEG monitoring has not been fully resolved. Additionally, this article describes canonical correlation analysis as a successful method for removing muscle artifacts.

One of the most important characteristics of ES is abnormal inter-channel synchronization or so-called coherency. To assess interchannel EEG synchronization, canonical correlation analysis [33], normalized cross-correlation and imaginary part of coherency or phase synchronization are used [34]. The main disadvantage of estimation coherence is the necessity to average it over time epochs and frequency ranges [35]. The study of short-term frequency synchronization of signals in two EEG channels by comparing their WT ridges frequencies during a previously selected by physicians is presented in ES [36].

We did not find in the literature any information on taking into account one of the most important feature of ESs—EEG interchannel synchronization for detecting ESs. So this article describes a new approach to the segmentation of the initial long-term clinical multichannel EEG monitoring data of patients with epilepsy into temporal fragments suspicious of an ES, to reduce the quantity of EEG fragments. This approach is based at first on the EEG WT ridges segmentation of the into frequency-synchronized fragments, and secondly with a thresholding of the ridge spectral power density.

Another part of this paper is devoted to a new approach to the diagnosis and assessment of rehabilitation of patients after traumatic brain injury (TBI). TBI is an insult to the brain from an external mechanical force, which can lead to permanent or temporary impairment of cognitive, physical, and psychosocial functions. The most used EEG methods of investigation TBI are spectral analysis, absolute and relative amplitude and power, coherence, and symmetry between homologous pairs of electrodes (see review [36]). A multivariate support system has been developed to quantify and classify by Random Forest classifier TBI stage based on analysis of EEG power in various frequency ranges [37]. A study [38] investigated the possibility of detecting moderate TBI according to the Glasgow Coma Scale [39] by EEG amplitude analysis and convolutional neural network classification. Recently, a single channel system was developed for real-time mild TBI detection with Convolutional Neural Networks classifier of EEG power in different frequency ranges [40]. The proposed method can be applied for screening of the moderate TBI and for selection of the patients for further diagnostics and treatment. In [41] the analysis of the EEG data applying the energy, sample entropy, approximate entropy, Lempel–Ziv complexity features demonstrated the increase in sample entropy was related with the functional recovery, i.e., the rehabilitation dynamics of the injured brain region. EEG-based neurofeedback is used for cognitive rehabilitation of patients with TBI [42].

Our approach is based on the analysis of the network of phase sinchronized EEG channels WT ridges in patients with moderate TBI. The interchannel phase difference of the EEG is determined during cognitive tests at the points of the frequency-modulated wavelet spectra ridges. We investigate the neurons connectivity disruption of the brain after TBI and consider the inter-channel phase connectivity between EEG channels during cognitive tests. It does not depend on the EEG signal amplitude. In this paper Section 2 contains the basic formulae and conditions for their application. Section 3 describes a new approach to segmentation of long-term EEGs into temporal fragments suspicious of an ES by interchannel WT ridges frequency sincronization and power spectral density thresholding. Section 4 describes a new approach to determine the evaluation of rehabilitation positive dynamics of patients with moderate TBI.

## 2. Materials and Methods

We studied long-term (from several hours to several days) initial EEG records of preoperative patients with epilepsy, obtained in laboratory of invasive neurointerfaces of the Research Institute TechnoBioMed. A.I. Yevdokimov Moscow State University of Medicine and Dentistry. The segmentation method was used for several days’ 19-channel EEG. The records were carried out according to the 10–20 system [43] in reference montage with a sampling rate of 256 Hz. Power supply artifacts were removed from all EEG channels using a notch filter at frequencies multiples of 50 Hz. The use of Morlet WT in the frequency range from 0.5 to 22 Hz, so myographic artifacts were rejected.

The records of 19-channel EEG were considered, therefore the quantity of pairs of channels is 171 for the group of control volunteers (18 subjects) and for the group of patients with moderate TBI (12 subjects), where three patients had repeated EEG records after rehabilitation in two cognitive tests. Cognitive tests were calculation-logical (CT1) and spatial-pattern (CT2). During the CT1 test, the doctor randomly spoke words from the category of “clothing” or “food” to the subject. During the test, the subject counted in their mind the number of items belonging to one of these categories. At the end of the test, the subject announced the result of the number of items. On the CT2 test, the doctor named an arbitrary time. The subject had to represent the position of the hands-on-dial in accordance with the indicated time. If both clock hands were in the same half of the dial, he said “yes”, and if they were in different halves, he kept silent. Investigations of control volunteers and patients with moderate TBI were carried out at the National Medical Research Center for Neurosurgery named after Academician N.N. Burdenko. All subjects were right-handed and signed written consent to participate in the research in accordance with the provisions of the Helsinki Agreement. The rehabilitation was performed for 1–2 months. The time of the rehabilitation was 40–45 min two times a week. The criteria for the inclusion of patients in the investigation were the ability to stand on their own and the ability to follow the doctor’s instructions, and also the absence of hemiparesis and other neurological disorders. The international 10–20 system of the position of scalp electrodes was used for EEG record. The recording time for every test was 60 s. EEG recording was carried out both during the tests and without them. The sampling rate of the EEG was 250 Hz in the processing of EEG signals. The original signals were recorded with a high-pass filter with a cut-off frequency of 0.5 Hz, a low pass filter with a cut-off frequency of 70 Hz. Then, a notch filter at frequencies multiples of 50 Hz and a Butterworth filter were used. The signals were filtered by a fourth-order Butterworth bandpass filter with a bandwidth from 2 to 10 Hz. The EEG records were analyzed without selecting individual fragments of the signal. However, the removal of outliers in the EEG signals was done with the Huber’s X84 method [44].

We considered EEG as an analytical signal with time-varying amplitude and frequency. The analytic signal was first locally represented as a modulated oscillation, demodulated by its own instantaneous frequency, and then Taylor-expanded at each point in time. We represent this signal as the following function:(1)$$S\left(t\right)=A_{S}\left(t\right)\exp\left(i\Phi_{S}\left(t\right)\right),$$
where $A_{S}\left(t\right)$ is the amplitude and $\Phi_{S}\left(t\right)$ is the phase of the signal. Continuous wavelet transform of signal S(t) is represented as:(2)$$W\left(a,b\right)=\int_{-\infty }^{\infty }S\left(t\right)\psi_{a,b}^{*}\left(t\right)\;dt$$
(3)$$\psi_{a,b}\left(t\right)=\frac{1}{\sqrt{\left|a\right|}}\psi \left(\frac{t-b}{a}\right)$$
where a,b,a≠0 are the real numbers defining the scale and the shift. We used the following function (Morlet mother wavelet) that was employed in the Matlab software:(4)$$\psi \left(t\right)=\frac{1}{\sqrt{\pi f_{b}}}\exp\left(-t^{2}/f_{b}\right)\exp\left(2\pi if_{c}t\right)$$
where $f_{b}$ is a positive and related with the variance of Gaussian function and $f_{c}$ is a positive value that corresponds with central frequency. The Morlet wavelet transform can be represented as follows:(5)W(a,b)=M(a,b)exp(iΦ(a,b)),
where M(a,b) is the absolute value of wavelet transform and Φ(a,b) is the phase of wavelet transform (Equation 2).

Substituting expressions (Equation 3) and (Equation 4) in formula (Equation 2), we obtain:(6)$$W\left(a,b\right)=\frac{1}{\sqrt{a\pi f_{b}}}\int_{-\infty }^{\infty }A_{S}\left(t\right)\exp\left(-\frac{{\left(t-b\right)}^{2}}{a^{2}f_{b}}\right)\exp\left(i\left(\Phi_{S}\left(t\right)-2\pi f_{c}\frac{t-b}{a}\right)\right)\;dt,$$
usually (in Matlab) $f_{b}=f_{c}=1$.

Integral (Equation 6) is approximately calculated by the method of stationary phase [45]. Under certain conditions, the main contribution to the integral is made by the imaginary part of the exponential function, since the contributions of rapidly changing phases cancel each other out, and the contribution is made by the values located at the point of the stationary phase. The stationary phase method is applicable when the amplitude A(t) of the signal exhibits relatively slow changes compared to fast changes in the total signal associated with fast changes in phase, for example, and asymptotic properties are satisfied concerning the window ψ(t) under the following assumptions; so that the following conditions are satisfied [11]:(7)$$\left|\frac{\;d\Phi_{S}\left(t\right)}{\;dt}\right|\gg \left|\frac{1}{A_{S}\left(t\right)}\frac{\;dA_{S}\left(t\right)}{\;dt}\right|,\left|\frac{1}{A_{S}\left(t\right)}\frac{\;dA_{S}\left(t\right)}{\;dt}\right|\ll \left|\frac{1}{\psi (t)}\frac{\;d\psi (t)}{\;dt}\right|$$

The relationship between the phase from expression (Equation 5) and the phase from expression (Equation 6) is given as follows:(8)$$\Phi \left(t\right)=\Phi_{S}\left(t\right)-2\pi f_{c}\left(\frac{t-b}{a}\right)$$

For the stationary phase Φ(t), we have
(9)$$\frac{\;d\Phi (t)}{\;dt}=\Phi_{S}^{\prime }\left(t\right)-\frac{2\pi f_{c}}{a}=0$$

Such a condition is satisfied at t=t(a). To estimate the integral from formula (Equation 6), we expanded the phase Φ(t) in a Taylor series up to a polynomial of the second degree in the neighborhood of point t=t(a) till the order ${\left(t-t\left(a\right)\right)}^{2}$:(10)$$\Phi \left(t\right)\approx \Phi_{S}\left(t\left(a\right)\right)-2\pi f_{c}\left(\frac{t(a)-b}{a}\right)+\frac{1}{2}\Phi_{S}^{\prime \prime }\left(t\left(a\right)\right){\left(t-t\left(a\right)\right)}^{2}$$

Below, we use the notation $\Phi_{S}\equiv \Phi_{S}\left(t\left(a\right)\right)$ and $\Phi_{S}^{\prime \prime }\equiv \Phi_{S}^{\prime \prime }\left(t\left(a\right)\right)$.

After substituting formula (Equation 10) into formula (Equation 6), we obtained an approximate value for the phase and absolute value of the wavelet transform:(11)$$\Phi \left(a,b\right)\approx \Phi_{S}-2\pi f_{c}\left(\frac{t(a)-b}{a}\right)+\frac{1}{2}\arctan\left(\frac{a^{2}f_{b}}{2}\Phi_{S}^{\prime \prime }\right)+\frac{2{\left(t\left(a\right)-b\right)}^{2}\Phi_{S}^{\prime \prime }}{4+a^{4}f_{b}^{2}{\left(\Phi_{S}^{\prime \prime }\right)}^{2}}$$
(12)$$M\left(a,b\right)\approx A_{S}\left(t\left(a\right)\right){\left(1+\frac{a^{4}f_{b}^{2}}{4}{\left(\Phi_{S}^{\prime \prime }t\left(a\right)\right)}^{2}\right)}^{-\frac{1}{4}}\exp\left(-\frac{{\left(t\left(a\right)-b\right)}^{2}a^{2}f_{b}{\left(\Phi_{S}^{\prime \prime }\right)}^{2}}{4+a^{4}f_{b}^{2}{\left(\Phi_{S}^{\prime \prime }\right)}^{2}}\right)$$

Expression (Equation 12) shows that the maximum of wavelet transform absolute value was reached at b=t(a). The instantaneous frequency at the ridge point $f_{r}$ at time moment t=t(a) was calculated using expression (Equation 9):(13)$$f_{r}\left(t\left(a\right)\right)=2\pi \frac{f_{c}}{a}$$

In this case, the maximal value of the wavelet transform (ridge) of the signal is given by
(14)$$\max_{a}\left|W(a,b)\right|\approx A_{S}\left(t\left(a\right)\right){\left(1+\frac{a^{4}f_{b}^{2}}{4}{\left(\Phi_{S}^{\prime \prime }t\left(a\right)\right)}^{2}\right)}^{-\frac{1}{4}}$$
and the phase is approximated of a ridge point as
(15)$$\Phi \left(a,b\right)\approx \Phi_{S}+\frac{1}{2}\arctan\left(\frac{a^{2}f_{b}}{2}\Phi_{S}^{\prime \prime }\right)$$

As in [46], the relationship between the frequency of Fourier spectrum of the wavelet transform and the scales *a* of the wavelet transform (Equation 2) is given as follows:(16)$$f=\frac{f_{0}}{2a}+\frac{\sqrt{2+4{\left(\pi f_{0}\right)}^{2}}}{4\pi a}≅\frac{f_{0}}{a}=\frac{1}{a}$$
where $f_{0}$ is a wavelet central frequency and it is considered that $4{\left(\pi f_{0}\right)}^{2}\gg 2$. So, for the ridge points $\left(f_{r},t\right)$ we have:(17)$$\begin{array}{c} W_{r}\left(t\right)=\max_{f}\left|W(f,t)\right|,\;f_{r}\left(t\right)=\underset{f}{\arg\;\max}\left|W(f,t)\right|,\\ \Phi_{S}\left(t\right)≅\Phi_{r}\left(f,t\right)=\arctan\left(\frac{Im(W\left(t,f_{r}\right))}{Re(W\left(t,f_{r}\right))}\right), \end{array}$$
when the condition
(18)$$\frac{\Phi_{S}^{\prime \prime }}{2f_{r}^{2}}=\frac{f_{r}^{\prime }}{2f_{r}^{2}}\ll 1,$$
is satisfied.

Summarizing, it should be noted that, in contrast to other works, the obtained simple method for determining the ridge points as the maximum of the modulus of the wavelet spectrum at each time point was undoubtedly easy to calculate.

## 3. EEG Segmentation

This chapter describes an EEG segmentation method based on the study of Morlet wavelet transform ridges, which allows finding time intervals of interest in ES detection, which is used to analyze continuous long-term EEG monitoring data during post-processing. The long-term EEG segmentation method consists of the following stages, as shown in Figure 1: 1. signal filtration at frequencies multiples 50 Hz, 2. wavelet Morlet transform of signals, 3. determination of wavelet spectrogram ridges, 4. marking time intervals with interchannel synchronization, 5. marking time intervals with power spectral density (PSD) values above the threshold, 6. intersections of time intervals, 7. visualization of a segmented signal with marked time intervals.

Let us show how segmentation was carried out using the example of an EEG recording fragment containing an ES. For each EEG channel, we calculated the wavelet spectrogram (2) and the ridges of the wavelet spectrogram (17) in frequency range [0.5; 22] Hz.

Generalized ESs were characterized by changes in power in several EEG channels and the synchronization of different channels pairs. In order to estimate the inter-channel synchronization, the modulus of the frequency difference at the points of the ridges was calculated for each pair of channels. If the modulus of the difference was less than ε, then there was synchronization $Sync_{i,j}$, otherwise, it was not:(19)$$Sync_{i,j}\left(k\right)=\left\{\begin{array}{c} 1,|f_{ri}\left(k\right)-f_{rj}\left(k\right)|\leq \varepsilon \\ 0,|f_{ri}\left(k\right)-f_{rj}\left(k\right)|>\varepsilon \end{array}\right.$$
where $f_{ri}$, $f_{rj}$ are the frequencies of the ridges of the wavelet spectrograms on the *i* and *j* EEG channels, *k* is the point of the ridge.

Figure 2 shows the projection of the wavelet spectrogram onto the PSD-frequency plane of the sinusoidal signals with a frequency of 2 and 2.5 Hz. At ε=0.5 the peaks were distinguishable. On smaller epsilons, the peaks could merge.

Nearby points at which condition (Equation 19) was satisfied were combined into fragments. Fragments between which the time interval was less than 10 s were combined into one. For each fragment, the beginning and end times of synchronization in pairs of channels were calculated. Table 1 shows a histogram of the number of synchronized fragments depending on the duration in 19 pairs of EEG channels. For neurophysiological considerations, this work considered fragments with a duration of 10 s or more.

Figure 3 shows an example of ES fragment from observed EEG recording with visualization of the presence of synchronization in channels pairs.

As can be seen, EEG synchronization could be observed in not all channels simultaneously (Figure 3). In this example, synchronization was observed in most channel pairs from about 5970 s to 6000 s, but synchronization began to appear earlier in a smaller number of pairs.

The time intervals in which the inter-channel synchronization in the frequency of the ridges was recorded could correspond to both ES and artifacts of chewing, sleep, and random physical influences on the electrodes, which generated artifacts of a non-epileptic nature.

A characteristic feature of the ES was a sharp change in the amplitude over a short period of time. Therefore, in addition to searching for time intervals in which there was synchronization on several pairs of EEG derivations, the detection of areas with high values of the power spectral density (PSD) was carried out. In order to understand the idea of the method, let us consider the histogram of the ridge points of the wavelet spectrogram, calculated for one of the leads. The peak of the histogram contained about $1.2\times 10^{6}$ points, the maximum PSD value at which the number of ridge points tended to 0, about $2.8\times 10^{7}$ μ V${}^{2}$/Hz. Such a histogram gave a large peak in the region of low PSD values and did not allow us to estimate the distribution of the ridge points, therefore, Figure 4 shows the “window” of the PSD histogram. As can be seen from the figure, the number of ridge points with low PSD values was large and could be interpreted as noise. It was necessary to separate the informative points of the ridge from the noise.

In order to separate the points of the ridge of the wavelet spectrogram related to high-amplitude electrical activity from noise, it was required to find the threshold value of PSD Tr. The ridge $PSD_{r}$ values of the wavelet spectrogram were determined as follows:(20)$$PSD_{r}\left(t\right)=\left\{\begin{array}{c} PSD_{r}\left(t\right),PSD_{r}\left(t\right)\geq Tr\\ 0,PSD_{r}\left(t\right)<Tr \end{array}\right.$$

The points of the ridge $PSD_{r}\left(t\right)$ lying between the nearest points $PSD_{r}\left(t\right)=0$ was called the ridge segment. Figure 5 shows a histogram of the number of ridge segments from the PSD threshold Tr to $Tr=5\times 10^{5}$ μ V${}^{2}$/Hz. At large values of $PSD_{r}$, the number of segments tended to be 0. The $PSD_{r}$ values could differ greatly not only from patient to patient but also by channels; therefore, it was required to determine the threshold value adaptively. Figure 5 shows a sharp decrease in the number of fragments with an increase in the threshold value. To identify the threshold value, the second derivative of segments quantity from the threshold changes was analyzed. After reaching a local maximum (circle mark in Figure 5), it became negligible. This means that the segments quantity linearly decreased with growing threshold. In Figure 5 threshold PSD value $Tr=1.5\times 10^{5}$ μ V${}^{2}$/Hz. With this choice of the threshold, most ESs detected by the expert and a small number of artifacts like ES were observed. Figure 6 shows an example of a segmented ridge of a wavelet spectrum containing an ES.

Figure 7 shows a fragment of a daily EEG signal showing an ES. Marks of the expert neurophysiologist are green vertical lines; the blue line, repeating the waveform, marks the fragment on which the synchronization was recorded on several pairs of EEG derivations; the dotted rectangles mark the areas found by the threshold method. Thus, the method of application for the search for ES is shown.

For the 5-hour EEG 2017 the overall synchronized fragments duty more than 10 were found (see Table 1), 112 segments were detected by thresholding, and finally we obtained nine intersected segments. with total duration total duration about 4 min. Earlier [47], we showed that by processing synchronous video of these nine fragments four fragments were recognized as moving artefacts. As a result, there were five segments left with a total duration of 4 min.

The detection of ES on EEG is complicated by the presence of many non-epileptic artifacts in signals received from scalp electrodes: electromyographic, motor, instrumental human actions, etc. An overview of various types of artifacts is given in [48]. Therefore, there is a need to develop methods to differentiate ES from artifacts of a non-epileptic nature.

To solve this problem, an algorithm was proposed, which consisted of studying the broadband peaks of the wavelet spectrograms, which were characteristic of an ES and a chewing artifact. Let us make a comparison using the example of wavelet spectra of an epileptic seizure (Figure 8) and chewing (Figure 9). We analyzed slices of wavelet spectrograms frequency $f_{cur}\left(t\right)$ higher ridge frequency $f_{r}\left(t\right)$, for example, at Figure 8 and Figure 9$f_{cur}\left(t\right)=4$ Hz (green line).

For each slice, we calculated Fourier spectra. Figure 10 Fourier spectra of ES and chewing artifact at $f_{cur}=4$ Hz. The frequency of the main peak and full width at half maximum (FWHM) of the Fourier spectrum were calculated.

Figure 10 shows differences between ES and chewing artifacts. The main peak frequency of the Fourier spectrum was at 0.71 Hz for the chewing artifact, and at 1.86 Hz for the ES. Two peaks could be observed in ES slice spectrum, this can be interpreted as the presence of spike-wave activity. There was a difference in FWHM of the peaks of the Fourier spectrum of the slices: for a chewing artifact, it was almost 2 times more than for an epileptic seizure. This may mean that the seizure period was more stable than chewing.

## 4. The Estimation of Inter-Channel EEG Phase Connectivity in Patients with TBI

Various methods of EEG phase coherence are used to estimate the connectivity of brain regions. Usually, the phase coherency of signals is used for the estimation of the inter-channel connectivity of EEG [33,49,50]. Coherency $Coh_{xy}\left(f\right)$ is defined by the normalized complex cross-correlation $C_{xy}$ of signals x(t) and y(t):(21)$$Coh_{xy}\left(f\right)=\left|<C_{xy}>\right|,C_{xy}=\frac{S_{xy}\left(f\right)}{{\left(|S_{xx}\left(f\right)||S_{yy}\left(f\right)|\right)}^{1/2}},$$
and a phase coherency is defined as |<exp(iΔΦ)>|, where |<•>| is an averaging [33].

In coherency analysis of non-stationary EEG, it is necessary to average exp(iΔΦ) over different time intervals (epochs), and it is the first problem. The presence of the peak in the histogram of the phase difference in different epochs determines the presence or the absence of the phase synchronization in the absence of a peak. In addition to this, $Coh_{xy}\left(f\right)$ is averaged in preliminary selected frequency bands that are specified using neurophysiological data. Usually, these bands correspond to the delta (2–4 Hz), theta (4–8 Hz), alpha (8–12 Hz), beta (12–25 Hz) EEG, and other rhythms, and this is the second problem. These disadvantages of the coherency analysis that leads to instability in the definition of the inter-channel EEG connectivity. The validity of the coherent analysis of non-stationary EEG signals is questioned [34].

Another method for the estimation of the phase connectivity is to determinate the analytical signal $x^{*}\left(t\right)=x\left(t\right)+iH\left(x\left(t\right)\right)$, where H(x(t)) is the Hilbert transform [51]. Then, the phase of signal $x^{*}\left(t\right)$ is calculated as the arccosine (arcsine) of the ratio of the real (imaginary) part $x^{*}\left(t\right)$ to its modulus [52]. The phase synchronization of two signals takes place when:(22)$$|\Delta \Phi_{x,y}\left(t\right)|\leq const,$$
where $\Delta \Phi_{x,y}\left(t\right)=n\Phi_{x}\left(t\right)-m\Phi_{y}\left(t\right)$, Φ is a phase of the signal; n,m are integers. Then, the angular frequency of the signal can be found by the phase differentiating with respect to time. Numerical differentiation in the presence of phase fluctuations is an unstable procedure. Additionally, the disadvantage of the approach associated with the calculation of analytical signals is that it is well applicable for narrowband signals and not good enough for broadband signals [53].

The paper describes the methods and results of determining the phase-connected pairs of EEG channels of patients with moderate TBI before and after rehabilitation, which can be used to estimate the dynamics of treatment and rehabilitation of patients. The method of the estimation of the inter-channel EEG phase synchronization at the points of the ridges $f_{r}\left(t_{i}\right)$ of their wavelet-spectrograms (Equation 6) is considered as an inverse task for the task of modeling ridges:(23)$$f_{r}\left(t_{i}\right)=\arg\left\{\max_{f\left(t_{i}\right)\in \left[1:25\mathrm{Hz}\right]}\left(|W(t_{i},f\left(t_{i}\right))|\right)\right\},$$
on the condition (Equation 18). $\Phi_{x}\left(t\right)≅\Phi_{r}\left(f,t\right)=\arctan\left(\frac{Im(W\left(t,f_{r}\right))}{Re(W\left(t,f_{r}\right))}\right)$ according to (Equation 17), when the condition (Equation 18) is satisfied.

However, the ridge $W_{r}\left(t\right)$ can be considered as the frequency-modulated signal. It is necessary to take an unmodulated oscillation [54]:(24)$$x=A_{0}\sin\left(\omega_{0}t+\Phi_{0}\right),$$
and enter a variable frequency $\omega =\omega_{0}+\Delta \omega \xi \left(t\right)=\omega \left(t\right)$, where ξ(t) is some unknown function, and ω(t) is known.

Then if $\Phi_{0}=0$: (25)$$x=A_{0}\sin\left(\omega_{0}t+\Delta \omega \int_{0}^{t}\xi \left(t\right)\;d\;t\right)=A_{0}\sin\left(\omega \left(t\right)t\right),$$

Then it is possible to estimate the phase of the ridge as [55]:(26)$$\Phi \left(t,f_{r}\right)=2\pi f_{r}\left(t\right)t,$$

Figure 11 represents two ridge frequencies of the Morlet wavelet transform for two EEG channels. Ridge points are points of the maximum power spectral density. Fp1 EEG channel is indicated by the blue line. Fp2 EEG channel is indicated by the red line. The abscissa is the time in seconds; the ordinate is the frequency in Hz.

The EEG frequencies coincided in some time fragments. The phase of the ridge could be estimated with the formula (Equation 26) if the ridge frequency was known.

The phases of the EEG signals were calculated and compared at the points of the ridges $\left(t_{i},f_{r}\right)$ of their wavelet spectrograms in EEG records both with cognitive tests and without tests. Then, the phase difference of two signals x(t) and y(t) in two EEG channels was calculated. Next, the normalized histogram of portions $\rho_{x,y}=n_{x,y}/N$ in different pairs of EEG channels was calculated, where $n_{x,y}$ is the quantity of reference points of ridges with $|\Delta \Phi_{x,y}\left(t\right)|<0.01\pi$, *N* is a total quantity of EEG signal reference points in the test.

Figure 12 represents the normalized histograms of portions of the phase difference at the ridge points of the wavelet spectrograms of two EEG channels for the case of a phase coupled pair of EEG channels Fp1-Fp2, which were obtained by two methods. Figure 12a demonstrates the first way based on the calculation of the phase according to the formula (Equation 26). Figure 12b demonstrates the second way based on the calculation of the phase according to the formula (Equation 17).

Figure 12a demonstrates that the histogram of portions of the phase difference at the points of the ridge of the wavelet spectrograms calculated by the first way (Equation 26) had a higher and sharper peak versus the second way (Equation 17) (Figure 12b). Below, we will calculate the phase by formula (Equation 26).

Let $A=\max(\rho_{x,y})$ be the maximum values of the histogram in the cognitive test and let $B=\max(\rho_{x,y})$ are the maximum values of the histogram in the EEG record without a test. It is convenient to consider the difference D=A−B, which was sorted in order to increase $\max(\rho_{x,y})$. Figure 13 demonstrates the dependence of *D* sorted in increasing order versus the numbers of a pair of EEG channels and its derivative for a healthy subject in the CT1 test.

Figure 13 shows that the curve of the graph appears at some point *D*. It is advisable to consider pairs of channels with numbers greater than at point sharp of increasing of derivative *D* (black point) as phase-connected pairs. Thus, phase-connected pairs of EEG channels were identified before and after rehabilitation of patients with moderate TBI.

Figure 14 shows a block diagram of the developed algorithm for the determination of phase-connected EEG channels. The developed algorithm for the determination of phase-connected EEG channels consisted of the following stages, as shown in Figure 14: 1. Preprocessing of signals. It was outlier removing; notch filter at frequencies multiples of 50 Hz; filtering of signals with a Butterworth filter; 2. Calculation of wavelet spectra and ridges; 3. Calculation of the ridges phase at each point of the wavelet spectra ridges. Calculation histograms of the phase difference portions $\left(\rho_{x,y}\right)$ in two channels for 171 channels pairs. The determination $\max(\rho_{x,y})$ for each channel pairs; 4. Calculation of the difference between $\max(\rho_{x,y})$ with cognitive test and without a test (*D*), sorting in in-creasing order of *D*. Calculation derivative (*D*); 5. The determination of phase connected EEG channels. If the derivative (*D*) sharply increased with the growing pair, the pairs with numbers greater than at the sharp point of the increasing derivative (*D*) were considered as a phase connected pairs. If the derivative (*D*) did not sharply increase, it was impossible to identify phase connected pairs.

Figure 15 demonstrates the phase-coupled pairs of EEG channels for seven healthy subjects during the EEG recording in the CT1 test.

Figure 15 represents that the frontal regions and interhemispheric connections are activated in cognitive tests (CT1). Interhemispheric connections and connections in the frontal cortex in control subjects are activated during CT1 test in accordance with published work [56]. However, each control subject and patient with TBI are characterized by different phase-connected pairs due to the individuality of each person during the CT1 test. Therefore, we considered phase-connected pairs individually for each subject.

Figure 16 demonstrates the phase-connected pairs of EEG channels for the seven control subjects during the EEG recording in the CT2 test.

Figure 16 represents that the frontal regions and interhemispheric connections were activated in cognitive tests (CT2). Interhemispheric connections and connections in the frontal cortex in control subjects were activated during CT2 test in accordance with published work [56]. However, each control subject and patient with TBI were characterized by different phase-connected pairs due to the individuality of each person during the CT2 test. Therefore, we considered phase-connected pairs individually for each subject.

Figure 17 demonstrates phase-connected pairs of EEG channels for three patients with TBI during the EEG recording in CT1 and CT2 tests.

Figure 17 shows that phase-connected pairs appeared more in the parietal and occipital regions, than in the interhemispheric and frontal cortex in patients with TBI during CT1 and CT2 tests.

Additionally, the dynamics of inter-channel EEG synchronization of three patients with TBI before and after the rehabilitation was also investigated. The phase-connected EEG pairs in patients before and after rehabilitation were compared with the phase-connected pairs of the control group for each test. If interhemispheric connections or connections in the frontal cortex were activated in patients, as in control subjects in cognitive tests (CT1 and CT2), it could be concluded that the cognitive function had positive dynamics.

Figure 18 demonstrates that the positive dynamics could be seen of the rehabilitation of a patient with TBI in the CT1 test. If interhemispheric connections or connections in the frontal cortex in the CT1 test appeared after rehabilitation, as in the control subjects, the positive dynamics of rehabilitation could be concluded.

Figure 19 demonstrates that the positive dynamics could be seen of the rehabilitation of patients with TBI in the CT2 test.

Figure 19 demonstrates that the positive dynamics could be seen of the rehabilitation of a patient with TBI in the CT2 test because interhemispheric connections or connections in the frontal cortex were activated in patients, as in control subjects. If interhemispheric connections or connections in the frontal cortex in the CT2 test appeared after rehabilitation, as in the control subjects, it the positive dynamics of rehabilitation could be concluded.

Let us consider an example of the lack of progress of the rehabilitation of a patient with TBI. Figure 20 demonstrates the dependence of *D* sorted in increasing order versus the numbers of pairs of EEG channels and its derivative for a patient with TBI.

Figure 20b shows that there was no sharp increase in the *D* derivative after the rehabilitation during the cognitive calculate-logical test, in contrast to Figure 20a. Thus, it was impossible to clearly determine the quantitative signs and identify phase-connected pairs by the suggested method. It could be concluded that there was no progress in rehabilitation during the cognitive calculation-logical test.

## 5. Conclusions

The paper presents an approach for segmenting long-term 19-channel EEG monitoring data. For the signals, the ridges of Morlet wavelet transform were calculated. Interchannel synchronization was used as a new feature of epileptic seizure. We also used the adaptive thresholding of the wavelet spectrogram ridges for signal segmentation. The intersection of the synchronized and the power spectral density intervals were obtained. As a result, the total duration of the fragments for analysis by the doctor was reduced by more than 60 times. It was shown that the frequency of the peak of the Fourier spectrum of the cutoff of the wavelet spectrogram at a frequency higher than the frequency of the ridge during an epileptic discharge was 2.5 times higher than the frequency of the Fourier peak corresponding to chewing. The Fourier peak full width at half maximum of the chewing artifact was 2 times larger than that of ES.

A comparison of the phases of EEG at the points of the Morlet wavelet spectrogram ridges were used for evaluation the EEG interchannel phase synchronization during cognitive tests in control subjects and patients with moderate TBI. Calculation-logical and spatial-pattern cognitive tests were used. Interhemispheric connections and connections in the frontal cortex in control subjects are initiated during the cognitive tests. The possibility of determining the positive dynamics of rehabilitation during the initialization of interhemispheric connections and connections in the frontal cortex of the brain or the absence of progress in rehabilitation has been shown.

## Figures and Tables

**Figure 1 sensors-21-05989-f001:**
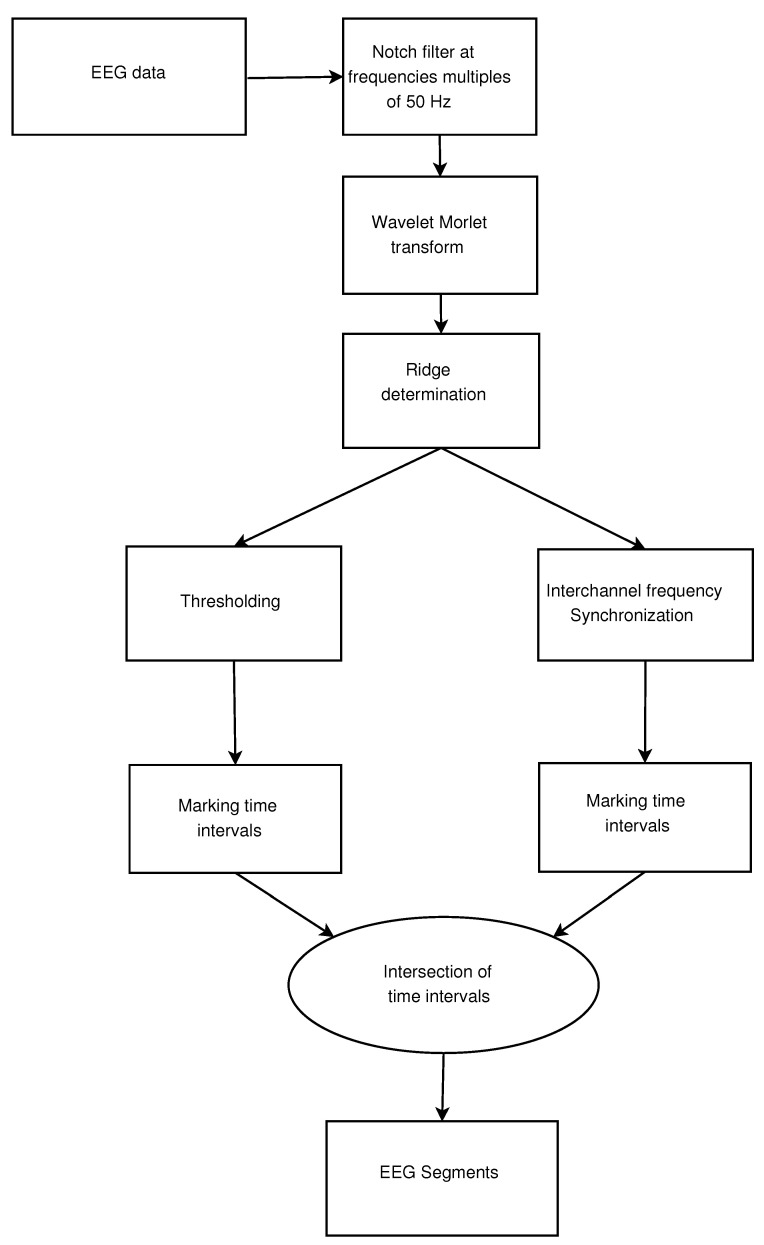
Block diagram of a long-term EEG segmentation.

**Figure 2 sensors-21-05989-f002:**
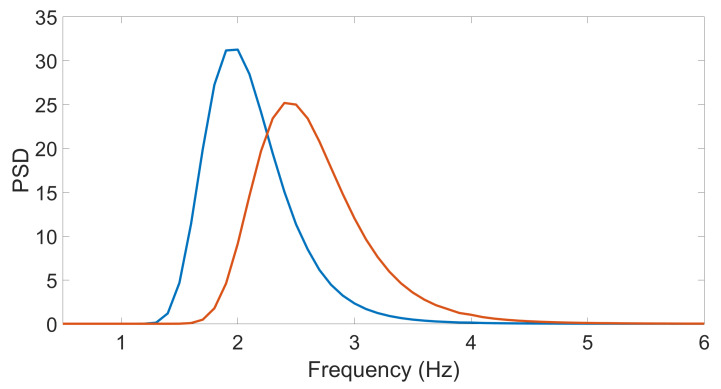
PSD-Frequency projection of wavelet spectrums of two sinusoidal signals: blue is 2 Hz, orange is 2.5 Hz.

**Figure 3 sensors-21-05989-f003:**
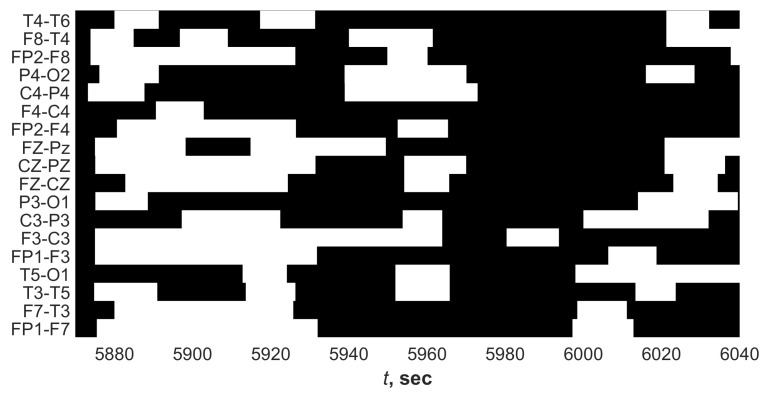
Fragment of daily EEG signal with ES illustrating the frequency of the wavelet spectrogram ridge synchronization in different pairs of EEG channels. Black shows the presence of synchronization. The ordinate shows the labels of channel pairs.

**Figure 4 sensors-21-05989-f004:**
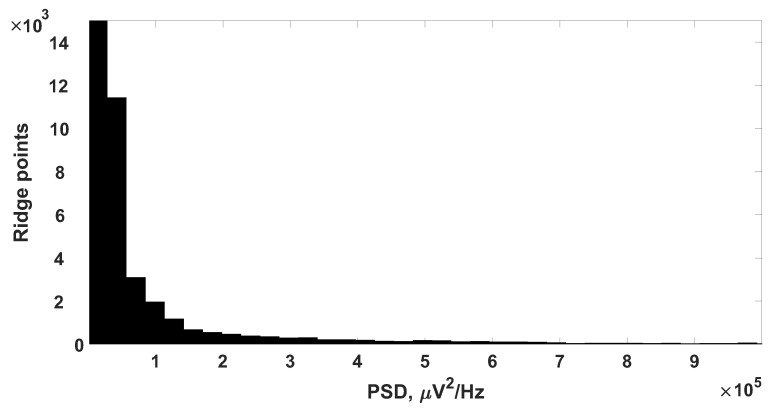
The window of the histogram of the PSD of the wavelet spectrogram ridge of the long-term EEG signal.

**Figure 5 sensors-21-05989-f005:**
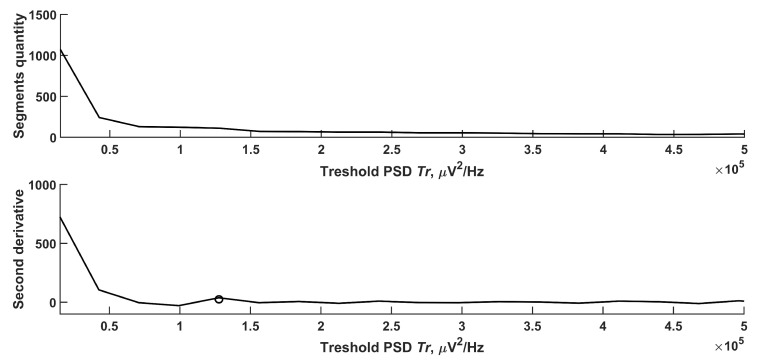
Histogram of the number of ridge segments from the threshold PSD Tr. The circle marks the local maxima at threshold value $Tr=1.27\times 10^{5}$ μ V${}^{2}$/Hz.

**Figure 6 sensors-21-05989-f006:**
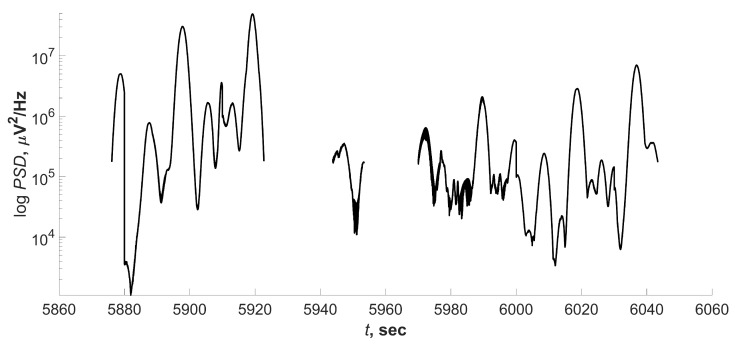
The segmented wavelet spectrogram ridge of the EEG signal fragment typical for ES. The upper figure shows a segmented ridge $PSD_{r}$, the bottom one shows $f_{r}$.

**Figure 7 sensors-21-05989-f007:**
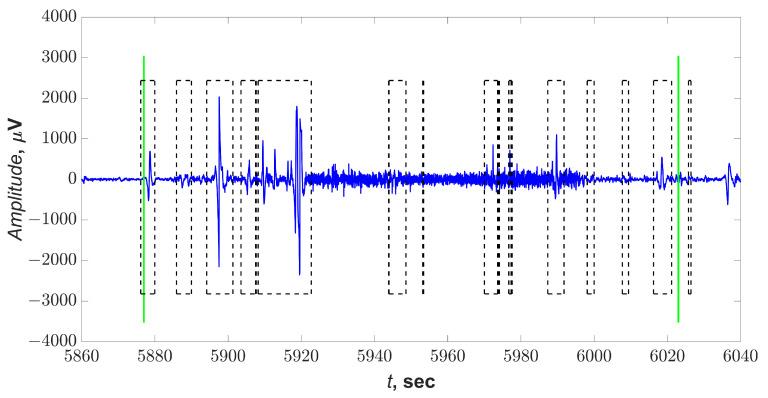
Fragment of an EEG with ES. Green vertical lines are expert marks. The blue line repeating the signal waveform is a mark obtained by searching for synchronized channel pairs. Dotted squares are marks obtained by the threshold method for detecting ES.

**Figure 8 sensors-21-05989-f008:**
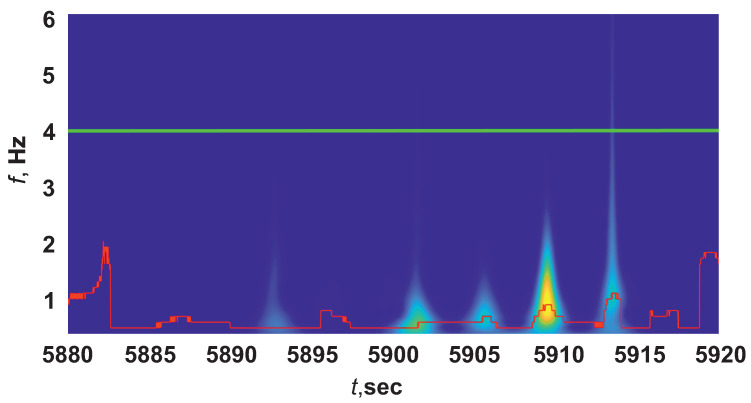
Wavelet spectrogram of an EEG with ES. Green line corresponds to slice of the wavelet spectrogram at the frequency $f_{cur}=4$ Hz.

**Figure 9 sensors-21-05989-f009:**
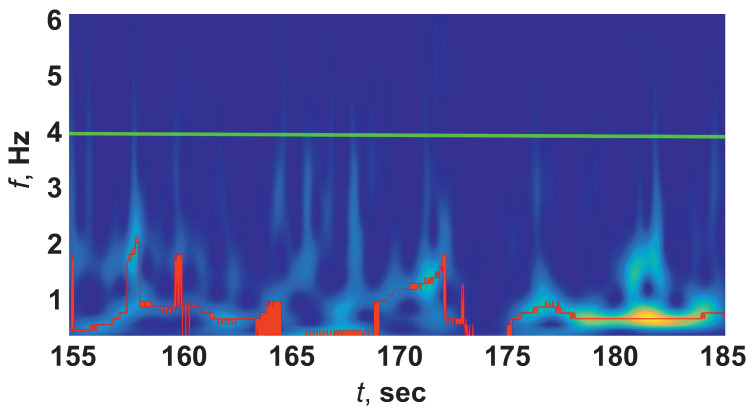
Wavelet spectrogram of an EEG with chewing artifact. Green line corresponds to slice of the wavelet spectrogram at the frequency $f_{cur}=4$ Hz.

**Figure 10 sensors-21-05989-f010:**
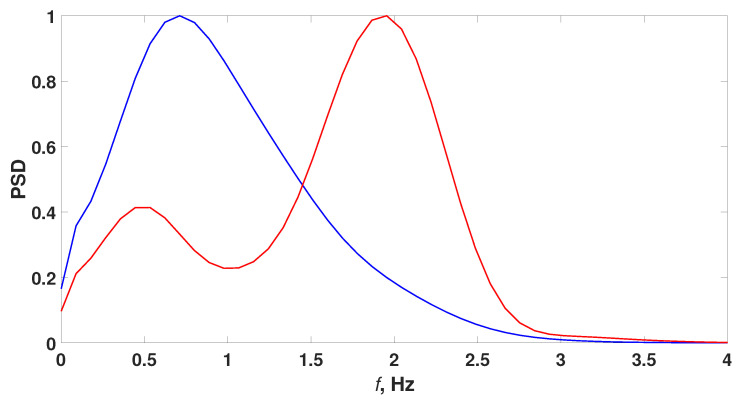
Fourier spectra of wavelet spectrograms slices $f_{cur}=4$ Hz. The red line is the Fourier Spectrum of the ES slice; the blue line is the Fourier spectrum of the chewing artifact slice.

**Figure 11 sensors-21-05989-f011:**
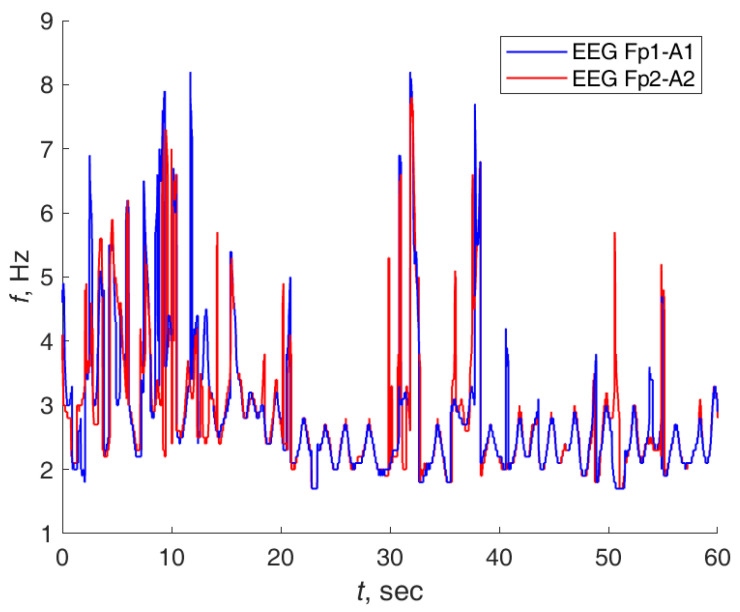
Ridge frequencies of the Morlet wavelet transform for two EEG channels. Fp1 EEG channel is indicated by the blue line. Fp2 EEG channel is indicated by the red line.

**Figure 12 sensors-21-05989-f012:**
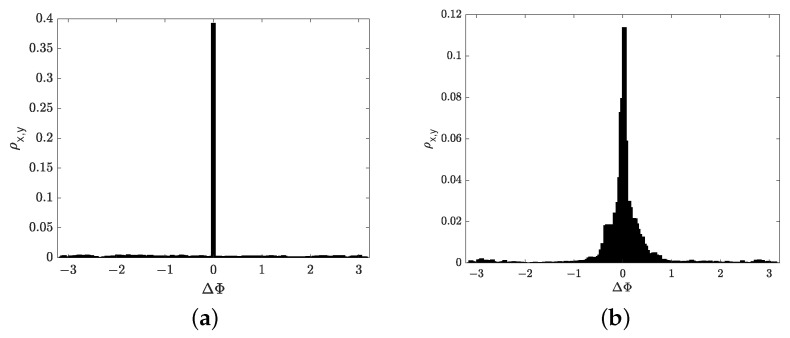
Histograms of portions $\rho_{x,y}$ for the phase difference at ridge points for two EEG channels. This example shows a phase-connected pair of Fp1-Fp2 EEG channels. (**a**) The histogram is obtained from the phase calculation by (Equation 26). (**b**) The histogram is obtained from the phase calculation by (Equation 17).

**Figure 13 sensors-21-05989-f013:**
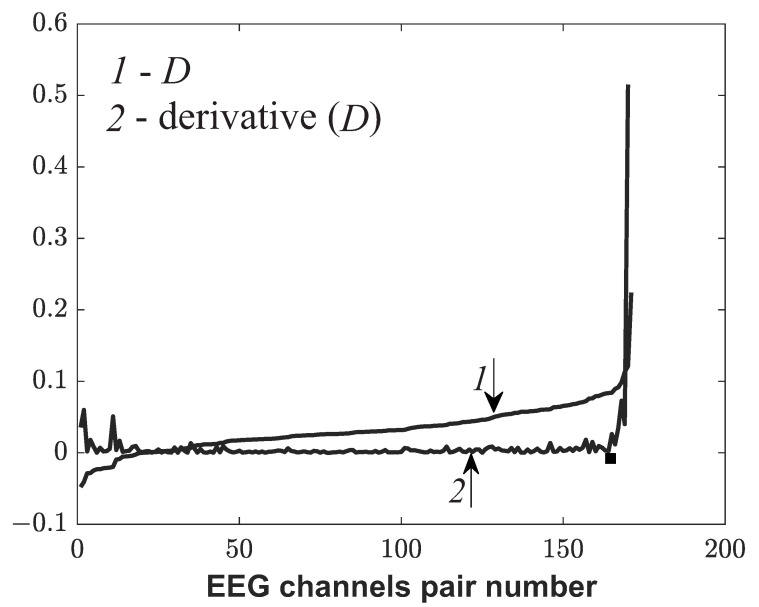
The dependence of *D* sorted in increasing order (line1) versus the numbers of a pair of EEG channels and its derivative (line 2) for a control subject in the CT1 test.

**Figure 14 sensors-21-05989-f014:**
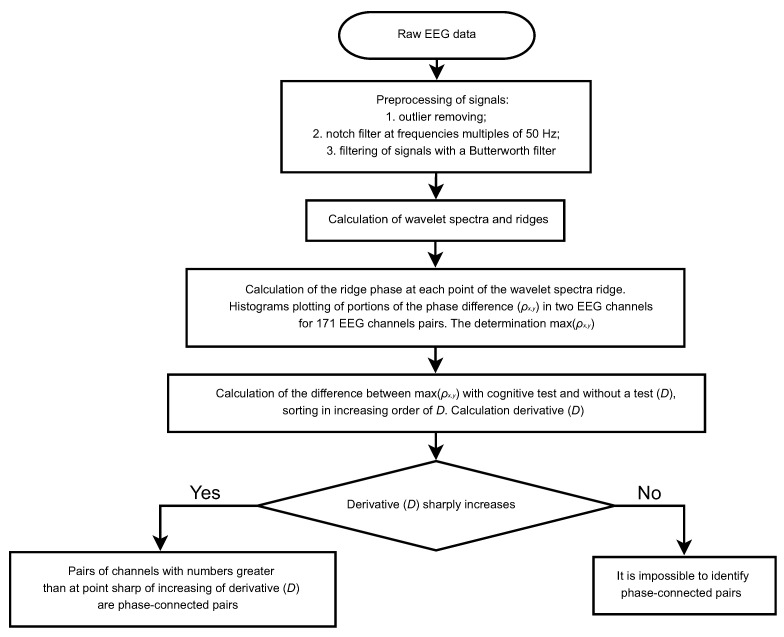
The block diagram of the developed algorithm for the determination of phase-connected EEG channels.

**Figure 15 sensors-21-05989-f015:**
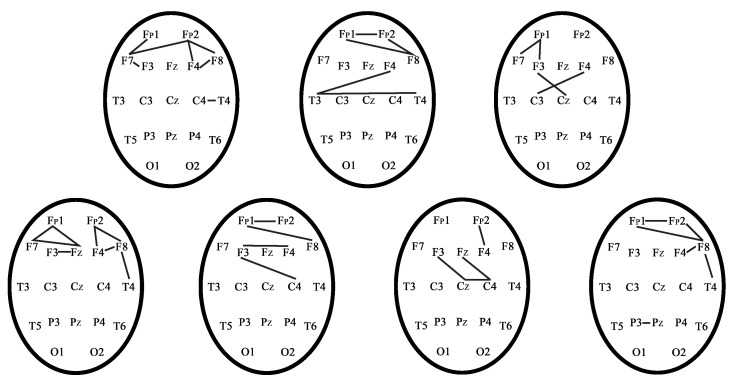
Phase-connected pairs of EEG channels of control subjects during the EEG recording in the CT1 test.

**Figure 16 sensors-21-05989-f016:**
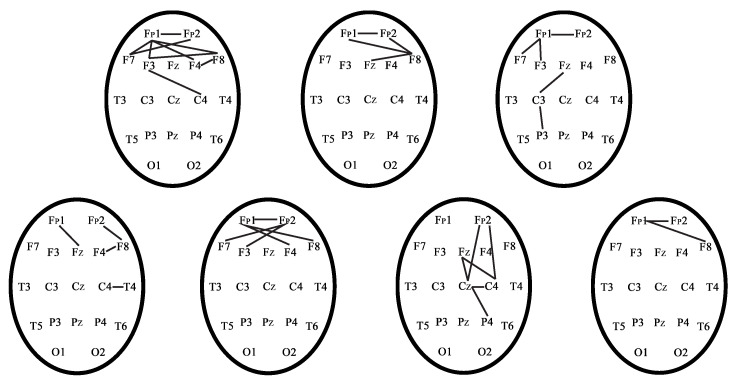
Phase-connected pairs of EEG channels of the control subjects during the EEG recording in the CT2 test.

**Figure 17 sensors-21-05989-f017:**
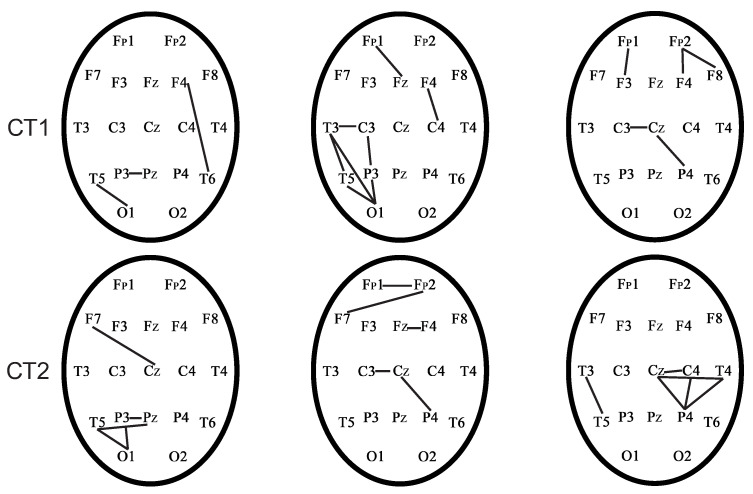
Phase-connected pairs of EEG channels of patients with TBI during the EEG recording in CT1 and CT2 tests.

**Figure 18 sensors-21-05989-f018:**
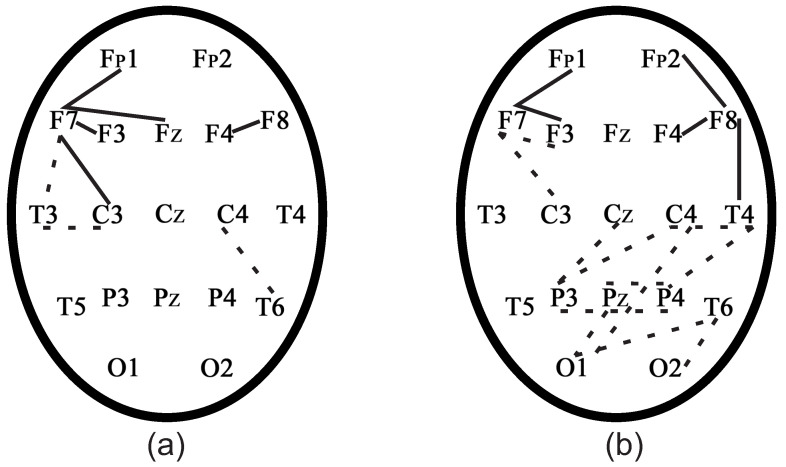
Phase-connected pairs of EEG channels of patients with TBI before (dotted lines) and after the rehabilitation (solid lines) in the CT1 test. (**a**) Patient 1. (**b**) Patient 2.

**Figure 19 sensors-21-05989-f019:**
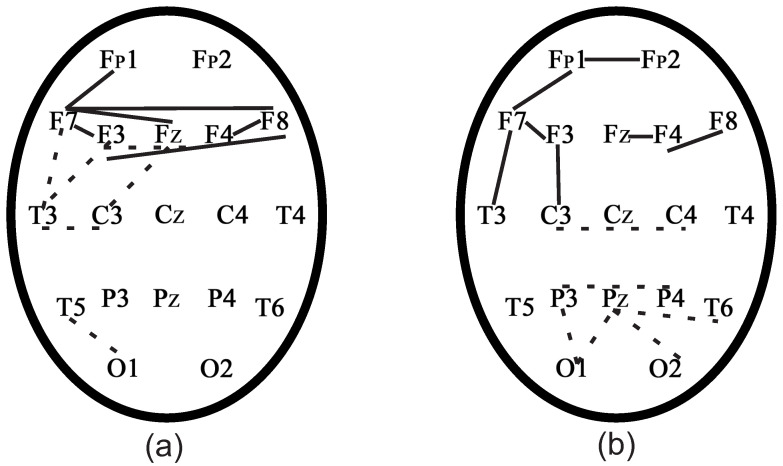
Phase-connected pairs of EEG channels of a patients with TBI before (dotted lines) and after the rehabilitation (solid lines) in the CT2 test. (**a**) Patient 1. (**b**) Patient 2.

**Figure 20 sensors-21-05989-f020:**
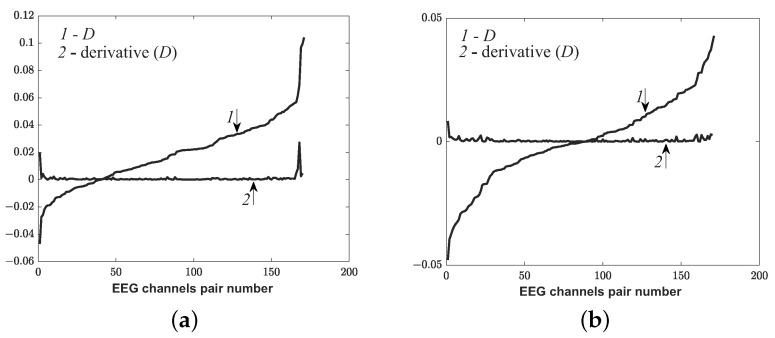
The dependence of *D* sorted in increasing order (line 1) versus the numbers of a pair of EEG channels and its derivative (line 2) for a patient with TBI in the CT1 test. (**a**) before the rehabilitation; (**b**) after the rehabilitation.

**Table 1 sensors-21-05989-t001:** Histogram of the number of synchronized fragments depending on the duration in 19 pairs of EEG derivations.

Pairs of Channels	Fragment Duration, s				
	>2	>5	>10	>15	>30
	≤5	≤10	≤15	≤30	
FP1-F7	3040	621	131	44	4
F7-T3	3683	430	84	29	9
T3-T5	4265	336	68	27	6
T5-O1	4210	375	77	40	4
FP1-F3	3413	534	90	20	1
F3-C3	3784	350	43	17	2
C3-P3	4140	397	41	15	1
P3-O1	4047	401	63	31	1
FZ-CZ	4250	362	35	18	1
CZ-PZ	4223	409	54	24	2
FZ-Pz	4044	329	31	16	0
FP2-F4	3352	535	99	46	7
F4-C4	3310	405	78	25	3
C4-P4	3869	421	75	15	1
P4-O2	4028	426	76	32	6
FP2-F8	3024	678	130	48	3
F8-T4	3385	518	88	46	5
T4-T6	3946	454	67	28	3
T6-O2	4190	470	76	28	3

## Data Availability

The data presented in this study are available on request from the corresponding author. The clinical data are not publicly available due to the ethical policy of the institute.

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
