# Peer review of "Wavelet Ridges in EEG Diagnostic Features Extraction: Epilepsy Long-Time Monitoring and Rehabilitation after Traumatic Brain Injuryâ€"

_sensors, 2021, doi:10.3390/s21185989_

Round 1
Reviewer 1 Report
In this paper, The authors presented the importance and application of wavelet transform ridges in the analysis of EEG for the diagnosis of epilepsy. The paper is not well structured and written. The following are my comments.
1) Abstract is very general. The abstract should be re-written by clearly mentioning the objectives, method obtained results and application of the study.
2) Page 5 line 158: typo. The Paper is written as pape.
3) The authors should add a figure showing the complete methodology used in this study.
4) It is very difficult to segregate the contents. The authors should use proper headings and workflow to show the methodology.
5) Following review article can be useful for preprocessing
Identification and removal of physiological artifacts from electroencephalogram signals: A review, 2018, IEEE ACCESS
6) Page 6, lines 206-207: How was the best result obtained? Any optimization method used?
7) The authors should more elaborate on the selection of Tr value.
8) Finally, The novelty and the contribution of the paper is not clear. What method is proposed? The results, literature and methods are mixed. The authors should provide a better-revised version for a clear understanding of what authors want to present to the scientific society.
Author Response
Dear Reviewers,
Thank you for reading our article and for your comments, which helped us to improve it.
We are trying to reply to your comments:
- Abstract is very general. The abstract should be re-written by clearly mentioning the objectives, method obtained results and application of the study.
We rewrote the abstract to indicate the objectives, approaches for solving investigated tasks, and obtained results,
- Page 5 line 158: typo. The Paper is written as pape.
We rechecked all words containing the word "pape" and corrected them to "paper" where it was appropriate.
- The authors should add a figure showing the complete methodology used in this study.
We add 2 block diagrams (figures 1 after line 226 and figure 14 after line 404) which show the methodology used in this study.
- It is very difficult to segregate the contents. The authors should use proper headings and workflow to show the methodology.
We have placed descriptions of the developed approaches into the "Introduction" section (lines 87-101, 121-131). We have also described investigated data and methods of their acquiring in the Materials and Methods section (lines 133-168).
- Following review article can be useful for preprocessing
Identification and removal of physiological artifacts from electroencephalogram signals: A review, 2018, IEEE ACCESS
We analyzed and added the information from this article in the revised introduction (lines 75-86).
- Page 6, lines 206-207: How was the best result obtained? Any optimization method used?
We didn’t use the optimization method. The choice of epsilon value 0.5 from the comparison of the separated wavelet spectra of the two sinusoidal signals in PSD-frequency projection. We provided figure 2 explaining the choice of epsilon (lines 237-239).
- The authors should more elaborate on the selection of Tr value.
We have added clarifications regarding the selection of the threshold (lines 277-283).
- Finally, The novelty and the contribution of the paper is not clear. What method is proposed? The results, literature and methods are mixed. The authors should provide a better-revised version for a clear understanding of what authors want to present to the scientific society.
The novelty of the article is due to the use of interchannel synchronization of the wavelet spectrograms ridges both in the segmentation of long-term baseline EEG in the diagnosis of epileptic seizures and in the assessment of the rehabilitation of patients with traumatic brain injury using the analysis of the network of phase-related EEG channels. It is reflected in block diagrams 1 and 14, the introduction (lines 87-101, 121-126), in the results of parts 3 and 4, and in the conclusion (lines 453-471). We provide a better-revised version of the article: Introduction, all parts and conclusion were revised also taking into account your comments.
We are attaching an updated version of the article.

Reviewer 2 Report
Dear authors ,
With interest we were reading the paper. We noticed that a new algorithm has been designed and applied in two applications. But we have the following critical comment:
- We miss the overall performance of the proposed algorithm. In the paper only some examples are presented.
- We expect that the algorithm will be benchmarked with other algorithms (see survey papers)
Guangda Liu, Ruolan Xiao, Lanyu Xu and Jing Cai* Minireview of Epilepsy Detection Techniques Based on Electroencephalogram Signals. Front. Syst. Neurosci., 20 May 2021 | https://doi.org/10.3389/fnsys.2021.685387
Sani Saminu 1,2,* , Guizhi Xu 1,*, Zhang Shuai 1 , Isselmou Abd El Kader 1 , Adamu Halilu Jabire 3 , Yusuf Kola Ahmed 2 , Ibrahim Abdullahi Karaye 1 and Isah Salim Ahmad 1. A Recent Investigation on Detection and Classification of Epileptic Seizure Techniques Using EEG Signal Brain Sci. 2021, 11, 668. https://doi.org/10.3390/brainsci11050668 -
In the second application the authors compare their work with an application using convolution neural network, reference [25]. We miss again a comparison of results.
Author Response
Dear reviewer,
Thank you very much for reviewing our article.
We reply to your comments:
- We miss the overall performance of the proposed algorithm. In the paper only some examples are presented.
We have included in the article a numerical estimation of the performance of the EEG segmentation algorithm (lines 289-294).
- We expect that the algorithm will be benchmarked with other algorithms (see survey papers)
Guangda Liu, Ruolan Xiao, Lanyu Xu and Jing Cai* Minireview of Epilepsy Detection Techniques Based on Electroencephalogram Signals. Front. Syst. Neurosci., 20 May 2021 | https://doi.org/10.3389/fnsys.2021.685387
Sani Saminu 1,2,* , Guizhi Xu 1,*, Zhang Shuai 1 , Isselmou Abd El Kader 1 , Adamu Halilu Jabire 3 , Yusuf Kola Ahmed 2 , Ibrahim Abdullahi Karaye 1 and Isah Salim Ahmad 1. A Recent Investigation on Detection and Classification of Epileptic Seizure Techniques Using EEG Signal Brain Sci. 2021, 11, 668. https://doi.org/10.3390/brainsci11050668
Thank you, we add these articles to the introduction. Comparison of our algorithm with other methods as in the presented reviews is not possible. The reason is that we are solving a different kind of problem. In our work, we provide a way to segment raw long-term EEG data, while the reviews present methods for classifying data, cleared of artifacts and marked by a physician as ictal and interictal fragments. The next reason is that the existing classification methods use single-channel processing methods and don’t consider interchannel synchronization. We are considering initial EEG data processing containing different artifacts. Our approach is based on a joint interchannel synchronization and power spectral density segmentation with the aim of reducing the overall duty of EEG fragments that must be analyzed by physicians.
- In the second application the authors compare their work with an application using convolution neural network, reference [25]. We miss again a comparison of results.
Comparison of our algorithm with the described method in the review is not possible. The reason is that we are solving a different kind of problem. In our work, we investigate the neuron’s connectivity disruption of the brain after TBI and consider the inter-channel phase connectivity between EEG channels during cognitive tests. Our approach allows to determine the positive dynamics of rehabilitation or the lack of progress in rehabilitation. It does not depend on the EEG signal amplitude. In [Chi Qin Lai et al. Detection of Moderate Traumatic Brain Injury from Resting-State Eye-Closed Electroencephalography Computational Intelligence and Neuroscience, Volume 2020, Article ID 8923906] the possibility of detecting moderate TBI according to the Glasgow Coma Scale by EEG amplitude analysis and convolutional neural network classification during Resting-State Eye-Closed is investigated. The next reason is that the described method does not consider interchannel phase EEG synchronization.
We are attaching an updated version of the article.

Round 2
Reviewer 1 Report
The authors have significantly improved the manuscript. All comments are addressed. However, there are minor typos. For example, in Figure 1, thresholding is written as tresholding, etc. The author should proofread the manuscript before final publication.
Reviewer 2 Report
We requested a benchmark of the proposed algorithm, but the authors state that the application of other authors are different. Then we don't understand why oter algorithms are not used for the same application. We realise that there this is some work to do postponed to the next future